# Knowledge, attitude, prevention practice and lived experience towards cutaneous leishmaniasis and associated factors among residents of Kutaber district, Northeast Ethiopia, 2022: A mixed method study

**Abebe Kassa Geto**[1]*, **Asmamaw Malede**[2], **Mistir Lingerew**[2], **Alebachew Bitew Abie**[2], **Gete Berihun**[2], **Ayechew Ademas**[2], **Leykun Berhanu**[2], **Genanew Mulugeta Kassaw**[3], **Belachew Tekleyohannes Wogayehu**[4], **Metadel Adane**[2]

1 Department of Nursing and Midwifery, Dessie Health Science College, Dessie, Ethiopia, 2 Department of Environmental Health, College of Medicine and Health Sciences, Wollo University, Dessie, Ethiopia, 3 Department of Public Health, College of Health Sciences, Woldia University, Woldia, Ethiopia, 4 Department of Environmental Health, Debre Birhan Health Science College, Debre Birhan, Ethiopia

* abebekassa2129@gmail.com

**Editor:** Álvaro Acosta-Serrano, University of Notre Dame, UNITED STATES OF AMERICA

## Abstract

### Background

Cutaneous leishmaniasis is a widespread parasitic infection in Ethiopia. Few studies have been conducted on knowledge, attitudes, and prevention practice related to cutaneous leishmaniasis, and the existing studies have been mainly without qualitative support. This study aimed to assess the knowledge, attitude, prevention practice and lived experience towards cutaneous leishmaniasis and associated factors among residents of the Kutaber district, Northeast Ethiopia.

### Methods

A convergent parallel mixed method was conducted among 636 residents (for quantitative) and 20 in-depth interview participants (for qualitative) of the Kutaber district from July 1 to August 15, 2022. Quantitative data were entered into Epi-Data version 4.6 and exported to SPSS version 25 for cleaning and analysis. ATLAS.ti software version 8.0 was used for the analysis of qualitative data.

### Results

The survey respondents in Kutaber district showed good knowledge (47.5%), a positive attitude towards (54.1%) and a good prevention practice (35.3%) regarding cutaneous leishmaniasis. Residents who were unable to read and write [AOR = 0.15] had lower odds to have good knowledge about cutaneous leishmaniasis. Residents aged >54.5 years [AOR = 0.33] had lower odds to have a positive attitude towards cutaneous leishmaniasis. Males

**Data Availability Statement:** All relevant data are in the manuscript and its supporting information files.

**Funding:** The author(s) received no specific funding for this work.

**Competing interests:** The authors have declared that no competing interests exist.

[AOR = 1.76] had a good prevention practice towards cutaneous leishmaniasis. Five main study themes were formed following the qualitative analysis of the data.

## Conclusion

Residents of Kutaber district have a poor overall knowledge and prevention practice towards cutaneous leishmaniasis, despite having a positive attitude. The educational status of residents and years of residence were factors significantly associated with knowledge about cutaneous leishmaniasis. Age and years of residence were factors significantly associated with attitude towards cutaneous leishmaniasis. Gender, age, and household wealth were factors significantly associated with prevention practice towards cutaneous leishmaniasis. Cutaneous leishmaniasis in Kutaber is a true health problem.

### Author summary

Cutaneous leishmaniasis (CL), a prevalent disease caused by an intracellular protozoan transmitted by a vector, is a serious public health issue, particularly in Kutaber district, where it causes significant physical and psychosocial problems among the residents. Even though it is not fatal, cutaneous leishmaniasis can lead to complications, lifelong scar and disfigurement, resulting in social isolation and discrimination. We showed that this is strongly associated with the district residents' poor knowledge and prevention practice towards the disease. Cutaneous leishmaniasis, a true health problem of the community, is worsened due to limited efforts of healthcare professionals, limited roles played by the media in health information communication, budget problem, low attention given from the government, lack of information dissemination, and lack of awareness. This implied that there is a need of rigorous provision of health education on the cause, transmission mechanism, intermediate host, breeding site, sandfly vector, treatment and implementation of prevention strategies.

## Introduction

Leishmaniases, which are caused by protozoan parasites from over 20 *Leishmania* species and transmitted to humans by infected female phlebotomine sandflies (*Phlebotomus* and *Lutzomyia*), have four main forms: Visceral Leishmaniasis (VL or kala-azar), Post-Kala-azar Dermal Leishmaniasis (PKDL), Cutaneous Leishmaniasis (CL), and Mucocutaneous Leishmaniasis (MCL) [1]. CL is a disease with clinical signs of skin or mucosal lesions with a parasitological confirmation of a positive smear or culture as a diagnosis [2]. It is the most common form of leishmaniasis disease in the world [1].

CL is a widespread parasitic infection caused by a single-celled flagellated parasite belonging to the genus *Leishmania* [3]. The findings of some studies showed that there was a poor level of knowledge about the cause, transmission, treatment, and preventive measures for the disease in the communities [4,5]. The disease CL is prevalent in many countries globally, with an estimated 12 million people affected and an annual incidence of 2–2.5 million cases. About 98 countries are affected, and around 350 million people are at risk of contracting the disease [6]. CL is more widely distributed, with about one-third of cases occurring in each of three epidemiological regions, the Americas, the Mediterranean basin, and western Asia from the

Middle East to Central Asia [7,8]. The ten countries with the highest estimated case counts, Afghanistan, Algeria, Colombia, Brazil, Iran, Syria, Ethiopia, North Sudan, Costa Rica, and Peru, together account for 70 to 75% of global estimated CL incidence [7].

A recent global burden analysis of CL listed 19 countries in Sub-Saharan Africa (SSA) as the top 50 high-burden countries [9]. CL is an endemic disease in East Africa, particularly in Ethiopia, which is among the countries with a high burden of CL [10,11]. In Ethiopia, CL has been known and described since 1913 by Martoglio who noted a vernacular name for the disease, indicating its familiarity to the people [12] and a more recent systematic review and meta-analysis revealed that the pooled CL prevalence in community-based cross-sectional studies was 20.13% [13].

The estimated number of cases per year ranges from 20,000 to 30,000, and the disease is mainly endemic in highland areas with an elevation of 1400 to 3175 m above sea level [11]. It is the most neglected disease in Ethiopia among all tropical diseases, causing skin lesions, mainly ulcers, which result in permanent scars, serious disability, stigmatization, and mental health problems for the community [14,15]. Moreover, CL causes disfigurement, leading to social and psychological impacts such as anxiety, depression, and low quality of life. This also affects economic productivity, making it a social, cultural, and health problem. Misconceptions about the disease have led to infected individuals being socially excluded, ranging from minor to severe physical and emotional isolation, which is attributed to a lack of knowledge about the disease [16]. In Ethiopia's Tigray region, people were more exposed to the disease due to the gaps in prevention methods for CL among the community. The main gaps in prevention methods reported were sleeping outdoors, working outside at night, being negligent on bed net utilization and failed to use chemical house spraying [17]. The disease is an overlooked public health problem in the Amhara region; which has the second highest pooled prevalence (23.62%) of CL next to Southern Nation Nationality and Peoples Region (SNNPR) (34.5%) in Ethiopia [13]. *Leishmania aethiopica* is the main cause of CL in the country [18], particularly in the Kutaber district where the disease is endemic [19].

Few studies have been conducted regarding knowledge, attitude, and prevention practice (KAP) towards CL and the findings on factors associated with KAP towards CL are very scanty. Almost all studies investigated so far were only quantitative and descriptive which were not supported by qualitative studies. Moreover, there was no epidemiological data that have been reported concerning the knowledge, attitude, prevention practice, and lived experience of residents about CL in the Kutaber district. Therefore, the present study aimed to determine the knowledge, attitude, prevention practice, lived experience, and associated factors toward CL among residents of the Kutaber district, Northeast Ethiopia. The results of this study will help policy-makers, healthcare planners, and other concerned bodies to plan and implement strategies to create awareness, prevent, and control CL.

## Materials and methods

### Ethics statement

An ethical clearance letter was obtained from the Ethical Review Committee (ERC) of Wollo University College of Medicine and Health Sciences, with reference number CMHS 1423/2014. Permission letters were taken from the Kutaber district health office and Boru-Meda hospital. Informed written consent was obtained from the study participants. The participant's right to refuse or withdraw from participating in the interview was fully maintained and the information provided by each participant was kept strictly confidential. The study was conducted based on the Declaration of Helsinki, 2008.

## Study area

The study was conducted in the Kutaber district, Northeast Ethiopia. It is located in the South Wollo Zone of the Amhara Regional State at a distance of 421 Kilometers (Km) from Addis Ababa, the country's capital. The district's capital is Kutaber, located on the main highway about 20 Km north of Dessie, the Zonal capital. It has 22 rural and 1 urban kebeles (the smallest administrative area in Ethiopia). In 2022, it had a total population of 112,130 with males figuring 57,612, and the total number of households was 26,077 [20].

## Study design and period

A convergent parallel mixed methods study was conducted from July 1 to August 15, 2022.

## Populations

**Source population.** **For the quantitative part**–All residents of the Kutaber district were the source population of the study.

**For the qualitative part**–Residents of Kutaber district who personally have experienced the disease CL and key informants of the district were the source population of the study.

## Study population

**For the quantitative study**–All residents of the selected kebeles in the Kutaber district were the study population of the study.

**For the qualitative study**–All residents of Kutaber district who personally were confirmed for having CL and key informants of the district were the study population of the study.

## Study unit

Individual was the study unit for both the quantitative and qualitative parts of the study.

## Eligibility criteria

**Inclusion criteria.** Those household heads or members who were legal residents (lived for 6 months or more) of the district and aged greater than or equal to 18 years were included in the quantitative part of this study. Community residents and key informants were included in this qualitative analysis of the study. For community residents the inclusion criteria were: being community residents and fulfilling both criteria (i.e., those who were legal residents of Kutaber district and residents of the district who personally were confirmed for having CL). For key informants, the inclusion criteria were: informants that were thought to give rich information about the phenomenon (i.e., CL) in the district and those who knew closely and had a direct involvement to that community (i.e., health professionals from South Wollo zone health department NTD office, district health office, healthcare professionals from Boru-Meda hospital and health centers of the district, health extension workers in the district, students, religious leaders, community elders of the district).

## Sample size determination

For the quantitative part of the study, the total sample size (n) was calculated using single population proportion formula with a 50.4% [21] proportion of good prevention practice. A 95% confidence level with a 5% margin of error, a 1.5 design effect and finally adding a 10% non-response rate, which yielded a total sample size of 636.

For the qualitative part of the study, a total of 20 participants (11 community residents that personally were confirmed for having the disease CL and 9 key informants) participated in an in-depth interview of the study and this was decided based on the point of information saturation.

## Sampling techniques

The selection of Kutaber district was done based on the occurrence of considerable CL cases registered at the dermatology department of Boru-Meda Hospital between 2016 and 2021 along with the ecology of the district. For the quantitative part of the study, a two-stage sampling was used to reach study participants. From a total of 23 "kebeles" in the Kutaber district, six "kebeles" were selected via a lottery method. Then after, a proportional allocation was employed to determine the sample size in each selected "kebeles" based on the number of households in each "Kebeles". Households from each selected "Kebele" were selected by computer-based simple random sampling technique (**Fig 1**).

For the qualitative part of the study, a purposive (criterion) sampling technique was used to select study participants for an in-depth interview.

## Study variables

The dependent variables of the study were knowledge about CL, attitude towards CL, and prevention practice towards CL. The independent variables of the study were socio-economic and demographic variables (gender, age, household size, educational status, occupation, marital status, household wealth category, place of residence, and years of residence), environmental variables (wall surface of the house made from, wall condition of the house, roof of house made from, floor of house made from, presence of latrine, location of the house from creeks/waterways and type of energy source used for cooking/heating) and behavioral variables (habit of dumping animal dung near the house, habit of filling cracks and animal burrows, habit of weeding round the home environment, habit of opening window at night, habit of working

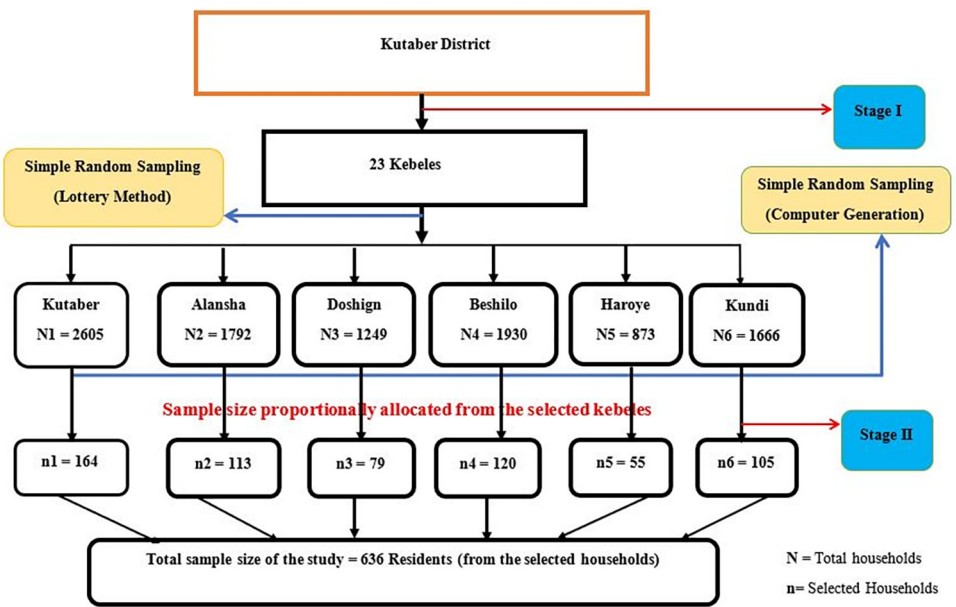

**Fig 1. Schematic presentation of the sampling procedure to select participants in Kutaber district, 2022.**

outside at night, habit of spending time near/at gorge, open defecation habit, habit of visiting traditional healers, media use, previous education about CL and knowing someone with CL).

## Operational definitions and measurements

**Cutaneous leishmaniasis.** is a disease with clinical signs of skin or mucosal lesions with a parasitological confirmation of a positive smear or culture as a diagnosis [2].

**Knowledge (overall)** about CL was measured using 14 item questions containing:

Identification of CL manifestation, ever got CL, the transmission of CL via the urine of bats, transmission of CL via sandfly bite, sign/s of CL, location of CL lesions/scars, habitat of the sandfly, communicability of CL, acquiring of CL in traveling, biting time of the vector, seriousness of the disease, preventability of CL, prevention measures for CL and curability from CL. Each of the questions has a score of 1 point for correct and 0 for incorrect and don't know answers/responses.

**Good Knowledge**: Participants who have scored greater than or equal to the mean score (8.4) of knowledge measurement questions.

**Poor Knowledge:** Participants who scored less than the mean score of knowledge measurement questions.

**Attitude (overall)** about CL was evaluated using 12 item questions comprising:

CL is a problem in the area, treatability of CL, the outcome of CL, effects due to the occurrence of CL, a season for a high incidence of CL, CL transmission via direct contact, the importance of environmental sanitation, feeling informed about CL, breeding places of the sandfly, spirituality of CL, a relation of CL with hyraxes and impression about CL. Attitude questions were designed with a five-point Likert scale [[1] Strongly disagree, [2] Disagree, [3] Neutral, [4] Agree, [5] Strongly agree] with a minimum of 12 points and a maximum of 60 points for each respondent.

**Positive Attitude:** Participants who scored greater than or equal to the mean score (30.86) of attitude measurement questions.

**Negative Attitude:** Participants who scored less than the mean score of attitude measurement questions.

**Prevention Practice (overall)** about CL was measured using 8-item questions containing:

Bed net use, work time preference, sleeping outdoors, repellent utilization, proper garbage disposal, indoor residual spray in the last 12 months, participation in CL control, and preference of treatment method for CL.

**Good Prevention Practice:** Participants who have scored greater than or equal to the mean score (5.00) of practice measurement questions.

**Poor Prevention Practice:** Participants who have scored less than the mean score of practice measurement questions.

**Lived Experience About CL:** this is an inclusive term for physical and psycho-social impacts, treatments, prevention, information dissemination, and communication experiences about CL experienced in the district.

**Self-Stigmatization:** isolation of him/herself from the community on own.

**Social Stigmatization:** is the isolation practiced by the community on a victim or a person with CL.

**Proper Garbage Disposal:** is the process of applying not throwing away garbage around the house and containing the garbage using storage containers, bags, or pits designed for it.

**Principal Component Analysis (PCA)** is a multivariate statistical technique used to reduce the number of variables in a data set into a smaller number of 'dimensions' [22].

**Household Wealth:** is an economic indicator for the level of households which is analyzed by PCA. In this study, household wealth was determined for urban and rural households

separately. It was measured by the nationally harmonized parameters of wealth indicators and categorized into five groups [poorest (lowest), poorer (second), middle, richer (fourth), and richest (highest)] [23].

## Data collection tools and techniques

For the quantitative part of the study, data were collected by face-to-face interview and observation (mainly for urban and rural wealth indicators and environmental factors) using an adapted [5,21,24] semi-structured and pretested questionnaire. The questionnaire was adapted in a way to fit with the socio-cultural aspects of the community, and comprised: socio-economic and demographic, environmental, behavioral, knowledge-related, attitude-related, and practice-related questions regarding CL. A total of six environmental health professionals (two as supervisors and four as data collectors) participated in the data collection process.

For the qualitative part of the study, interview-guided questions were used to collect data from the study participants. The in-depth interview was undertaken by two data collectors (AB and GM). During an in-depth interview, a probing method was used and audio recording was done using smartphones.

## Data quality assurance

For the quantitative part of the study, the questionnaire was prepared first in English and then translated into Amharic (mother language fluently spoken by all participants) by language experts for fieldwork purposes and back to English for consistency. Before data collection, a two-day training was given for supervisors and data collectors by the principal investigator. The reliability of the questionnaire was checked by Cronbach's alpha coefficient, and its validity was checked by a pre-test.

To ensure the quality of the qualitative data, proper designing of the interview guide was done and expert judgment (involved from multi-disciplinary professions) was applied. The probing method was also used to bring out important hidden ideas, views, and information from the study participants. The audio was recorded using two separate smartphones and a power bank was availed to charge them if their battery goes too low.

## Data processing and analysis

For the quantitative part of the study, raw data was entered into Epi-Data software version 4.6 and exported to Statistical Packages for Social Sciences (SPSS) version 25 statistical software for analysis. Descriptive statistics such as frequency distributions, cross-tabulations, and measures of central tendencies were calculated for dependent and independent variables.

Bi-variable and multi-variable analysis using binary logistic regression was performed. PCA was also performed to determine the wealth status of households.

For the qualitative part of the study, audio records were transcribed verbatim into Amharic language and then translated back to English by the principal investigator and language experts. Then, quoting, coding, and Thematic Analysis (TA) were performed using ATLAS.ti software version 8.0. Important specific views of the respondents with their sayings were selected and presented promptly.

## Results

### Socio-economic and demographic characteristics of study participants

From a total of 636 study participants in this study, 612 participants provided complete information with a response rate of 96.2%. From a total of 612 study participants, 377 (61.6%) were

**Table 1. Socio-economic and demographic characteristics of Kutaber district residents, July 1 to August 15, 2022.**

| Characteristics | Freq. (n) | Perce.(%) | Characteristics | Freq. (n) | Perc.(%) |
|---|---|---|---|---|---|
| Gender | | | **Occupation** | | |
| Male | 235 | 38.4 | Farmer | 216 | 35.3 |
| Female | 377 | 61.6 | House Wife | 226 | 36.9 |
| Age | | | Merchant | 56 | 9.2 |
| 18–24.5 | 62 | 10.1 | Government Employ | 46 | 7.5 |
| 24.5–34.5 | 177 | 28.9 | Student | 37 | 6.0 |
| 34.5–44.5 | 174 | 28.4 | Unemployed | 18 | 2.9 |
| 44.5–54.5 | 110 | 18.0 | Others | 13 | 2.1 |
| >54.5 | 89 | 14.5 | **Household Size** | | |
| Marital Status | | | 1–2 | 84 | 13.7 |
| Single | 67 | 10.9 | 3–5 | 391 | 63.9 |
| Married | 448 | 73.2 | > 5 | 137 | 22.4 |
| Divorced | 51 | 8.3 | **Place of Residence** | | |
| Widowed | 46 | 7.5 | Rural | 452 | 73.9 |
| Educational Status | | | Urban | 160 | 26.1 |
| Unable to read and write | 166 | 27.1 | **Household Wealth** | | |
| Able to read and write | 106 | 17.3 | Poorest | 118 | 19.3 |
| Primary education | 152 | 24.8 | Poorer | 127 | 20.8 |
| Secondary education &above | 188 | 30.7 | Middle | 126 | 20.6 |
| Years of Residence | | | Richer | 121 | 19.8 |
| 1–2 Year/s | 50 | 8.2 | Richest | 120 | 19.6 |
| > 2 Years | 562 | 91.8 | | | |

females. The majority, 448 (73.2%) of the study participants were married. Regarding educational status, secondary education and above comprised 188 (30.7%) of the study participants. About 562 (91.8%) of the participants were lived in the district for more than 2 years. One hundred twenty-seven (20.8%) of the participants were grouped under poorer household wealth status (**Table 1**).

## Environmental characteristics of study participants

About 511 (83.5%) of the study participants lived in a house with its wall surface made from mud. About two-thirds (66.7%) of the participants lived in a house with a wall surface have no cracks/holes. Six hundred five (98.9%) of the participants lived in a house with its roof was corrugated iron sheet. About 394 (64.4%) of the participants lived in a house with its floor pasted with cow dung/mud. About 580 (94.8%) of the respondents had a latrine. Three hundred fifty-six (58.2%) of the participants had a house that was not located close to creeks/waterways. About 534 (87.3%) of the participants used wood as a source of energy for cooking and heating purpose.

## Behavioral characteristics of study participants

Three hundred seventy-six (61.4%) of the study participants had the habit of dumping animal dung near their houses. About 510 (83.3%) of the participants had no habit of working outdoors at night and 556 (90.8%) of the participants had no custom of spending time near/at the gorge. Three hundred ninety-one (63.9%) of the participants were media users. About 500 (81.7%) of the participants had no habit of open defecation. More than three fourth, 489 (79.9%) of the participants knew someone with CL.

## Knowledge about cutaneous leishmaniasis

The finding of this study revealed that about 321(52.5%) of the study participants had poor knowledge about CL and the remaining 291 (47.5%) of the study participants had good knowledge about CL. From a total of 612 study participants, 522 (85.3%) of them correctly identified/recognized the disease after the picture of CL ("*Kunchir*" in local Amharic term) manifestation was shown to them. About 560 (91.5%) of the study participants had heard about the disease CL. Of those who had heard about CL, participants that have never got CL were 506 (90.4%). Regarding the transmission of CL through the urine of bats, the majority, 338 (60.4%) of the participants said, CL is transmitted via the urine of bats ("*Yelelit Wof Shint*" in local Amharic term). Four hundred twelve (73.6%) of the participants said, "I don't know" regarding CL transmission by the bite of a sandfly **(Table 2)**.

## Attitude towards cutaneous leishmaniasis

The finding of this study revealed that about 331(54.1%) of the study participants had a positive attitude toward CL and the remaining 281 (45.9%) of the study participants had a negative attitude towards CL. About 278 (45.4%) of the study participants agreed or strongly agreed that CL is a health problem in their area but, about 230 (37.6%) of the participants disagreed or strongly disagreed with this idea. About 247 (40.2%) of the participants disagreed that CL is transmitted via direct contact from person to person. About 172 (28.3%) of the participants disagreed that CL is a spiritual disease and 50 (8.2%) of the participants strongly disagreed with this idea. Nearly half, 296 (48.4%) of the study participants had no opinion about whether CL has a relation with rock hyraxes or not **(Table 3)**.

## Prevention practice towards cutaneous leishmaniasis

The finding of this study revealed that about 396 (64.7%) of the study participants had a poor prevention practice toward CL and the remaining 216 (35.3%) of the study participants had a good prevention practice toward CL. More than two-thirds, 429 (70.1%) of the study participants do not used bed nets. About 540 (88.2%) of the participant's work time preference was in day time. About 559 (91.3%) of the participants didn't sleep outdoors. Four hundred seventy-five (77.6%) of the participants haven't used repellents for CL prevention. More than half, 320 (52.3%) of the participants performed garbage disposal properly. About 479 (78.3%) of the participant's house has never been sprayed with insecticides in the last twelve months. Regarding the treatment methods, about 245(40%) of the study participants preferred to use modern medicine over traditional or a combination of modern and traditional treatment if they get the disease CL **(Table 4)**.

## Factors associated with KAP of residents towards CL

**Factors associated with knowledge of residents about CL.** The finding of this study revealed that, residents who were unable to read and write (AOR = 0.15, 95% CI: 0.088–0.269) and able to read and write only (AOR = 0.50, 95% CI: 0.280–0.875) were 85% and 50% less likely to have good knowledge about CL respectively as compared to those who were in secondary education and above. The variables: years of residence (AOR = 0.21, 95% CI: 0.090–0.466), household wealth (AOR = 0.31, 95% CI: 0.166–0.563), media use (AOR = 0.67, 95% CI: 0.452–0.985) and knowing someone with CL (AOR = 0.53, 95% CI: 0.330–0.837) were also significantly associated (at P value < 0.05) with knowledge about CL **(Table 5)**.

**Table 2. Knowledge About CL among Residents of Kutaber district, July 1 to August 15, 2022.**

| Items | Frequency(n) | Percent (%) |
|---|---|---|
| **Recognition of CL Manifestation Image (n = 612)** | | |
| Able to identify as CL | 522 | 85.3 |
| unable to identify as CL | 90 | 14.7 |
| **Have you heard about CL? (n = 612)** | | |
| Yes | 560 | 91.5 |
| No | 52 | 8.5 |
| **Have you ever got CL? (n = 560)** | | |
| Yes | 54 | 9.6 |
| No | 506 | 90.4 |
| **Is CL transmitted by the urine of bats? (n = 560)** | | |
| Yes | 338 | 60.4 |
| No | 42 | 7.5 |
| I don't know | 180 | 32.1 |
| **Is CL transmitted by the bite of Sandfly? (n = 560)** | | |
| Yes | 109 | 19.5 |
| No | 39 | 7.0 |
| I don't know | 412 | 73.6 |
| **Is skin lesion the sign of CL? (n = 560)** | | |
| Yes | 363 | 64.8 |
| No | 160 | 28.6 |
| I don't know | 37 | 6.6 |
| **Are face, forehead, nostril, arm, leg and ear the parts of the body for the location of CL lesion/scar? (n = 560)** | | |
| Yes | 490 | 87.5 |
| No | 42 | 7.5 |
| I don't know | 28 | 5.0 |
| **Are rock crevices, caves, rodent burrows, leaf litters and vegetation the habitats of sandfly? (n = 560)** | | |
| Yes | 217 | 38.8 |
| No | 38 | 6.8 |
| I don't know | 305 | 54.4 |
| **Is CL a disease transmitted from an infected person to a healthy person? (n = 560)** | | |
| Yes | 309 | 55.2 |
| No | 136 | 24.3 |
| I don't know | 115 | 20.5 |
| **Is there a possibility of acquiring CL in travelling to endemic areas? (n = 560)** | | |
| Yes | 315 | 56.3 |
| No | 138 | 24.6 |
| I don't know | 107 | 19.1 |
| **Are dawn and dusk the preferred biting times of the vector? (n = 560)** | | |
| Yes | 194 | 34.6 |
| No | 54 | 9.7 |
| I don't know | 312 | 55.7 |
| **Is CL a serious disease? (n = 560)** | | |
| Yes | 508 | 90.7 |
| No | 35 | 6.3 |
| I don't know | 17 | 3.0 |
| **Is CL preventable disease? (n = 560)** | | |

*(Continued)*

**Table 2.** (Continued)

| Items | Frequency(n) | Percent (%) |
|---|---|---|
| Yes | 396 | 70.7 |
| No | 164 | 29.3 |
| **Are health education, hygiene, and sanitation the prevention measures of CL? (n = 560)** | | |
| Yes | 355 | 63.4 |
| No | 128 | 22.9 |
| I don't know | 77 | 13.7 |
| **Is complete cure from CL possible? (n = 560)** | | |
| Yes | 436 | 77.9 |
| No | 83 | 14.8 |
| I don't know | 41 | 7.3 |
| **Overall, Knowledge About CL** | | |
| **Poor** | **321** | **52.5** |
| **Good** | **291** | **47.5** |

## Factors associated with the attitude of residents towards CL

The finding of this study showed that, residents aged > 54.5 years were 67% less likely to have a positive attitude towards CL as compared to those who were aged 18–24.5 years (AOR = 0.33, 95% CI: 0.122–0.877). The variables: years of residence (AOR = 2.64, 95% CI: 1.325–5.263) and habit of visiting traditional healers (AOR = 0.38, 95% CI: 0.255–0.553) were also significantly associated (at P value < 0.05) with attitude towards CL (**Table 6**).

## Factors associated with prevention practice of residents towards CL

The finding of this study revealed that, Males were 1.76 times more likely to have a good prevention practice towards CL as compared to females (AOR = 1.76, 95% CI: 1.078–2.889). The variables: age [24.5–34.5 years (AOR = 0.16, 95% CI: 0.064–0.420) and 34.5–44.5 years (AOR = 0.32, 95% CI: 0.124–0.848)], household wealth [poorest (AOR = 0.36, 95% CI: 0.192–0.663), poorer (AOR = 0.47, 95% CI: 0.266–0.844) and middle (AOR = 0.44, 95% CI: 0.246–0.792)], condition of the wall surface of the house [cracked (AOR = 0.50, 95% CI: 0.322–0.789) and hole-formed (AOR = 0.29, 95% CI: 0.108–0.759)], location of the house from creeks/waterways (AOR = 0.63, 95% CI: 0.433–0.929), habit of open defecation (AOR = 0.51, 95% CI: 0.301–0.857) and knowing someone with CL (AOR = 0.58, 95% CI: 0.355–0.948) were also significantly associated (at P value < 0.05) with prevention practice towards CL (**Table 7**).

## Lived experience of residents towards cutaneous leishmaniasis

**Characteristics of in-depth interview study participants.**   Thirteen male and seven female participants were involved in the interview. Age ranged between 18 and 56 years with an average of 31.3 (±10.95) years (**Table 8**).

**Main study themes and sub-themes.**   Following the analysis of the qualitative data, five main study themes were formed based on the closeness and similarity of concepts/patterns.

## Theme 1: physical and psycho-social impacts of the disease

Overall, participants reported that the physical, social, and psychological impact on their life was significant. Their fear related to social stigmatization, the associated pain, failure in healing, disease complication, disability, permanent scar, and future re-infection was expressed by their speech and emotional inputs.

**Table 3. Attitude Towards CL among Residents of Kutaber district, July 1 to August 15, 2022 (n = 612).**

| Items | Measurements for attitude towards CL | | | | | | | | | |
|---|---|---|---|---|---|---|---|---|---|---|
| | Strongly Agree | | Agree | | Neutral | | Disagree | | Strongly Disagree | |
| | n | % | n | % | n | % | n | % | n | % |
| CL is a health problem in the area | 95 | 15.5 | 183 | 29.9 | 104 | 17 | 224 | 36.6 | 6 | 1.0 |
| Believing as CL can be treated | 92 | 15.0 | 409 | 66.8 | 52 | 8.5 | 56 | 9.2 | 3 | 0.5 |
| Disability is the outcome of CL if not treated early | 229 | 37.4 | 362 | 59.2 | 7 | 1.1 | 13 | 2.1 | 1 | 0.2 |
| The occurrence of CL in one member of the family affects the economy of the whole family | 92 | 15 | 448 | 73.2 | 18 | 2.9 | 53 | 8.7 | 1 | 0.2 |
| Autumn is the season at which the incidence of CL is at its peak/high | 21 | 3.4 | 164 | 26.8 | 295 | 48.2 | 129 | 21.1 | 3 | 0.5 |
| Believing as CL is transmitted by direct contact from person to person | 18 | 2.9 | 147 | 24 | 174 | 28.4 | 247 | 40.4 | 26 | 4.2 |
| Environmental sanitation is important for prevention of CL transmission | 48 | 7.8 | 414 | 67.6 | 100 | 16.3 | 46 | 7.5 | 4 | 0.7 |
| You feel you are well informed about CL | 15 | 2.5 | 122 | 19.9 | 183 | 29.9 | 278 | 45.4 | 14 | 2.3 |
| Vegetation area, rock cracks, termite piles and animal manures are the major breeding places of sandfly | 30 | 4.9 | 253 | 41.3 | 271 | 44.3 | 55 | 9 | 3 | 0.5 |
| CL is spiritual disease | 130 | 21.2 | 91 | 14.9 | 169 | 27.6 | 172 | 28.1 | 50 | 8.2 |
| CL has a relation with rock hyraxes | 42 | 6.9 | 136 | 22.2 | 296 | 48.4 | 127 | 20.8 | 11 | 1.8 |
| Worrying is the impression of the disease CL | 219 | 35.8 | 330 | 53.9 | 19 | 3.1 | 40 | 6.5 | 4 | 0.7 |
| **Overall Attitude Towards CL** | | | | | | | | | | |
| **Negative** | Frequency (n) | | 281 | | | | | | | |
| | Percentage (%) | | 45.9 | | | | | | | |
| **Positive** | Frequency (n) | | 331 | | | | | | | |
| | Percentage (%) | | 54.1 | | | | | | | |

**Physical impacts.**  CL had an impact on the body of victims including the pain and permanent scar which reduces the physical beauty.

*'There is no worst disease other than cutaneous leishmaniasis. It is hard. The disease lefts you with scars. The scar is harsh. Beyond the pain, its scar is not fully recovered. The scar decreases the beauty of the people.'* Stated a young 30-year housewife regarding the disease and its scar.

**Stigmatizations.**  According to the participants, stigma is the most frequently listed problems related to CL in the study community (**Fig 2**).

**Social stigmatization.**  Social stigmatization was frequently linked with *cutaneous leishmaniasis* lesions for different reasons such as the community's thinking as the disease is transmittable from person to person, through direct observation, via food, and through houseflies. A 25-year-old religious teacher described the reason for social stigma as: *'The first measure that the community takes is stigmatization. This is as a result of fearing the contact.'* A 24 -year-old female rural health center head also stated, *'Stigmatization is seen in the areas as they (community members) think housefly bring their (the patients') blood and transmit the disease to them.'* A 34- year- old rural merchant stated *'The first thing that the community stigmatizes the victim of CL is due to the thinking of the disease's communicability to them. Some people think, if they are close to him/her (the victim), they might get infected.'*

Most participants experiencing the disease face social stigmatization in their community with some of them were also stated, there was a rejection from their families too. A 27-year unemployed rural young man stated, *'. . . By that time, both my male and female friends left me and isolated themselves from me. (. . .) I have a girlfriend at the time and she loved me and I*

**Table 4. Prevention Practice Towards CL among Residents of Kutaber district, July 1 to August 15, 2022 (n = 612).**

| Items | Frequency (n) | Percentage (%) |
|---|---|---|
| **Used bed nets** | | |
| Yes | 183 | 29.9 |
| No | 429 | 70.1 |
| **Work time preference** | | |
| Day time | 540 | 88.2 |
| Night | 26 | 4.2 |
| Both day and night | 46 | 7.5 |
| **Sleeping Outdoor** | | |
| Yes | **53** | **8.7** |
| No | **559** | **91.3** |
| **Use of repellents for CL Prevention** | | |
| Yes | 137 | 22.4 |
| No | 475 | 77.6 |
| **Properly performing garbage disposal** | | |
| Yes | 320 | 52.3 |
| No | 292 | 47.7 |
| **Indoor residual spray in the last 12 months** | | |
| Yes | 133 | 21.7 |
| No | 479 | 78.3 |
| **Ever participated in CL control activities** | | |
| Yes | 13 | 2.1 |
| No | 599 | 97.9 |
| **CL treatment method Preference** | | |
| Modern Treatment | 245 | 40 |
| Traditional Treatment | 238 | 38.9 |
| Both Modern and Traditional Treatments | 129 | 21.1 |
| **Overall Prevention Practice Towards CL** | | |
| **Poor** | **396** | **64.7** |
| **Good** | **216** | **35.3** |

*loved her too before I was with cutaneous leishmaniasis. While I was with cutaneous leishmaniasis, she left me alone and every one said to me "You are not yourself." (. . .) The reason for why this has happened is that there was stigma from the community and there is a lack of character and motivation to strengthen you.'* Moreover, an 18- year male student described the stigma experienced in the community as: *'. . . For those who are severely affected and their nose is strongly affected by cutaneous leishmaniasis, people would prefer not to get close and play with them.'* Similarly, an 18-year female student stated the stigma practiced in the community as: *'. . . Because of the scar already they have, people don't need to be named as you are his friend or you are her friend.*

**Self- stigmatization (self- isolation).** According to the participants, victims isolate themselves from the community as a result of their fear related to social stigma and psychological feelings posed to them. A 27-year rural youth stated that, *'. . . I was alone throughout the day from morning to night without contact with anyone for about three* [3] *months. I slept alone and I ate alone.'* CL victims' interaction with the community is minimized due to their fear associated with the scar. An 18-year female student stated *'I didn't get out of my home because I am afraid due to the scar of cutaneous leishmaniasis.'* A 25-year religious teacher stated, *'It is not*

**Table 5. Bivariable and multivariable Binary Logistic Regression Analysis Result for Factors Associated with Knowledge about Cutaneous Leishmaniasis among Residents of Kutaber District, Northeast Ethiopia, July 1 to August 15, 2022 (n = 612).**

| Variables | Knowledge about CL | | COR (95% CI) | AOR (95% CI) |
|---|---|---|---|---|
| | Good | Poor | | |
| **Age** | | | | |
| 18–24.5 | 30 | 32 | 0.561(0.312–1.009) | **Ref.** |
| 24.5–34.5 | 61 | 116 | 1.375(0.769–2.460) | 0.562(0.241–1.309) |
| 34.5–44.5 | 98 | 76 | 1.484(0.794–2.775) | 1.495(0.612–3.653) |
| 44.5–54.5 | 64 | 46 | 0.795(0.414–1.525) | 2.402(0.920–6.266) |
| >54.5 | 38 | 51 | 0.561(0.312–1.009) | 1.112(0.414–2.987) |
| **Educational Status** | | | | |
| Unable to read and write | 41 | 125 | 0.240(0.152–0.379) | **0.153(0.088–0.269)\*** |
| Able to read and write only | 48 | 58 | 0.605(0.375–0.978) | **0.495(0.280–0.875)\*** |
| Primary Education | 94 | 59 | 1.165(0.754–1.802) | 1.118(0.674–1.854) |
| Secondary Education and Above | 108 | 79 | **Ref.** | **Ref.** |
| **Years of Residence** | | | | |
| 1–2 Year/s | 9 | 41 | 0.218(0.104–0.457) | **0.205(0.090–0.466)\*** |
| > 2 Years | 282 | 280 | **Ref.** | **Ref.** |
| **Occupation** | | | | |
| Government Employ | 21 | 25 | **Ref.** | **Ref.** |
| House Wife | 110 | 116 | 1.129(0.598–2.133) | 1.430(0.676–3.026) |
| Merchant | 23 | 33 | 0.830(0.378–1.823) | 0.666(0.272–1.630) |
| Farmer | 110 | 106 | 1.235(0.652–2.340) | 1.395(0.648–3.003) |
| Student | 17 | 20 | 1.012(0.424_2.412) | 1.035(0.316_3.390) |
| Unemployed | 7 | 11 | 0.758(0.249–2.301) | 0.887(0.253–3.107) |
| Others | 3 | 10 | 0.357(0.087–1.470) | 0.410(0.086–1.963) |
| **Household Wealth** | | | | |
| Poorest | 37 | 82 | 0.401(0.236–0.681) | **0.305(0.166–0.563)\*** |
| Poorer | 52 | 75 | 0.616(0.372–1.021) | 0.569(0.316–1.024) |
| Middle | 66 | 60 | 0.978(0.592–1.615) | 0.740(0.412–1.328) |
| Richer | 73 | 48 | 1.352(0.810–2.256) | 1.032(0.569–1.872) |
| Richest | 63 | 56 | **Ref.** | **Ref.** |
| **Habit of visiting Traditional healer** | | | | |
| Yes | 72 | 98 | 0.748(0.524–1.069) | 0.701(0.459–1.071) |
| No | 219 | 223 | **Ref.** | **Ref.** |
| **Media Use** | | | | |
| Yes | 200 | 191 | **Ref.** | **Ref.** |
| No | 91 | 130 | 0.669(0.479–0.933) | **0.667(0.452–0.985)\*** |
| **Previous Education about CL** | | | | |
| Yes | 11 | 6 | **Ref.** | **Ref.** |
| No | 280 | 315 | 0.485(0.177–1.328) | 0.707(0.233–2.140) |
| **Knowing Someone with CL** | | | | |
| Yes | 11 | 6 | **Ref.** | **Ref.** |
| No | 280 | 315 | 0.570(0.379–0.857) | **0.526(0.330–0.837)\*** |

**AOR** = Adjusted Odds Ratio, **COR** = Crude Odds Ratio, **Ref. = Reference,** \* = Significant association at P-value < 0.05, Hosmer and Lemeshow goodness of fit test (P = 0.378)

**Table 6. Bivariable and multivariable Binary Logistic Regression Analysis Result for Factors Associated with Attitude towards Cutaneous Leishmaniasis among Residents of Kutaber District, Northeast Ethiopia, July 1 to August 15, 2022 (n = 612).**

| Variables | Attitude towards CL | | COR (95% CI) | AOR (95% CI) |
|---|---|---|---|---|
| | Positive | Negative | | |
| **Gender** | | | | |
| Male | 137 | 98 | 1.319(0.949–1.832) | 1.438(0.922–2.243) |
| Female | 194 | 183 | Ref. | Ref. |
| **Age** | | | | |
| 18–24.5 | 40 | 22 | Ref. | Ref. |
| 24.5–34.5 | 97 | 80 | 0.667(0.367–1.213) | 0.560(0.239–1.312) |
| 34.5–44.5 | 88 | 86 | 0.563(0.309–1.025) | 0.447(0.180–1.106) |
| 44.5–54.5 | 63 | 47 | 0.765(0.402–1.456) | 0.559(0.218–1.431) |
| >54.5 | 43 | 46 | 0.491(0.252–0.957) | **0.327(0.122–0.877)\*** |
| **Marital Status** | | | | |
| Single | 42 | 25 | Ref. | Ref. |
| Married | 236 | 212 | 0.663(0.391–1.124) | 0.805(0.351–1.846) |
| Divorced | 25 | 26 | 0.572(0.273–1.199) | 0.728(0.261–2.026) |
| Widowed | 28 | 18 | 0.926(0.428–2.003) | 1.294(0.441–3.798) |
| **Years of Residence** | | | | |
| 1–2 Year/s | 37 | 13 | 2.594(1.350–4.986) | **2.641(1.325–5.263)\*** |
| > 2 Years | 294 | 268 | Ref. | Ref. |
| **Occupation** | | | | |
| Government Employ | 27 | 19 | Ref. | Ref. |
| House Wife | 113 | 113 | 0.704(0.370–1.338) | 1.162(0.559–2.417) |
| Merchant | 32 | 24 | 0.938(0.426–2.068) | 1.019(0.444–2.338) |
| Farmer | 118 | 98 | 0.847(0.444–1.615) | 1.472(0.703–3.081) |
| Student | 22 | 15 | 1.032(0.428–2.489) | 0.616(0.170_2.238) |
| Unemployed | 10 | 8 | 0.880(0.293–2.641) | 0.807(0.230–2.830) |
| Others | 9 | 4 | 1.583(0.425–5.903) | 1.559(0.397–6.127) |
| **Place of Residence** | | | | |
| Rural | 233 | 219 | 0.673(0.466–0.972) | 0.685(0.445–1.053) |
| Urban | 98 | 62 | Ref. | Ref. |
| **Household Wealth** | | | | |
| Poorest | 71 | 48 | 1.148(0.686–1.922) | 1.114(0.638–1.945) |
| Poorer | 69 | 58 | 0.923(0.558–1.527) | 0.975(0.569–1.672) |
| Middle | 57 | 69 | 0.641(0.387–1.062) | 0.694(0.404–1.193) |
| Richer | 67 | 54 | 0.963(0.578–1.603) | 1.017(0.591–1.751) |
| Richest | 67 | 52 | Ref. | Ref. |
| **Habit of Visiting Traditional Healer** | | | | |
| Yes | 64 | 106 | 2.527(1.756–3.637) | **0.375(0.255–0.553)\*** |
| No | 267 | 175 | Ref. | Ref. |
| **Previous Education about CL** | | | | |
| Yes | 7 | 10 | Ref. | Ref. |
| No | 324 | 271 | 1.708(0.641–4.547) | 1.294(0.448–3.735) |

**Ref. = Reference,** \* = Significant association at P-value < 0.05, Hosmer and Lemeshow goodness of fit test (P = 0.169), **AOR** = Adjusted Odds Ratio, **COR** = Crude Odds Ratio

**Others:** Carpenter [4], Daily laborer [4], Security guard [2], Driver [1], Hairdresser [1] and Mason [1]

**Table 7. Bivariable and multivariable Binary Logistic Regression Analysis Result for Factors Associated with Prevention Practice towards Cutaneous Leishmaniasis among Residents of Kutaber District, Northeast Ethiopia, July 1 to August 15, 2022 (n = 612).**

| Variables | Prevention Practice towards CL | | COR (95% CI) | AOR (95% CI) |
|---|---|---|---|---|
| | Good | Poor | | |
| **Gender** | | | | |
| Male | 95 | 140 | 1.436(1.023–2.014) | **1.764(1.078–2.889)\*** |
| Female | 121 | 256 | Ref. | Ref. |
| **Age** | | | | |
| 18–24.5 | 27 | 35 | Ref. | Ref. |
| 24.5–34.5 | 39 | 138 | 0.366(0.198–0.678) | **0.164(0.064–0.420)\*** |
| 34.5–44.5 | 63 | 111 | 0.736(0.408–1.327) | **0.324(0.124–0.848)\*** |
| 44.5–54.5 | 40 | 70 | 0.741(0.393–1.397) | 0.369(0.134–1.010) |
| >54.5 | 47 | 42 | 1.451(0.756–2.785) | 0.678(0.240–1.910) |
| **Educational Status** | | | | |
| Unable to read and write | 58 | 108 | 1.165(0.748–1.816) | 1.215(0.693–2.130) |
| Able to read and write | 44 | 62 | 1.540(0.939–2.524) | 1.780(0.991–3.196) |
| Primary Education | 55 | 98 | 1.218(0.775–1.913) | 1.561(0.925–2.633) |
| Secondary Education and Above | 59 | 128 | 1.165(0.748–1.816) | Ref. |
| **Occupation** | | | | |
| Government Employ | 18 | 28 | Ref. | Ref. |
| House Wife | 81 | 145 | 0.869(0.453–1.667) | 1.165(0.503–2.701) |
| Merchant | 18 | 38 | 0.737(0.326–1.666) | 0.601(0.233–1.552) |
| Farmer | 81 | 135 | 0.933(0.486–1.793) | 0.834(0.374–1.860) |
| Student | 12 | 25 | 0.747(0.301–1.851) | 0.347(0.093–1.297) |
| Unemployed | 3 | 15 | 0.311(0.079–1.229) | 0.265(0.051–1.394) |
| Others (*a*) | 3 | 10 | 0.467(0.113–1.930) | 0.514(0.097–2.731) |
| **Household Wealth** | | | | |
| Poorest | 29 | 90 | 0.328(0.189–0.569) | **0.357(0.192–0.663)\*** |
| Poorer | 39 | 88 | 0.451(0.268–0.759) | **0.474(0.266–0.844)\*** |
| Middle | 43 | 83 | 0.527(0.315–0.881) | **0.442(0.246–0.792)\*** |
| Richer | 46 | 75 | 0.624(0.373–1.043) | 0.616(0.346–1.097) |
| Richest | 59 | 60 | Ref. | Ref. |
| **Condition of wall Surface of the house** | | | | |
| No holes/cracks formed | 165 | 243 | Ref. | Ref. |
| Cracks formed | 45 | 126 | 0.526(0.355–0.780) | **0.504(0.322–0.789)\*** |
| Holes formed | 6 | 27 | 0.327(0.132–0.810) | **0.287(0.108–0.759)\*** |
| **Location of the house from creeks/waterways** | | | | |
| Close | 75 | 181 | 0.632(0.448–0.890) | **0.634(0.433–0.929)\*** |
| Not close | 141 | 215 | Ref. | Ref. |
| **Source of Energy for Cooking/heating** | | | | |
| Electricity | 26 | 31 | Ref. | Ref. |
| Wood and/Charcoal | 184 | 350 | 0.627(0.361–1.087) | 0.542(0.280–1.048) |
| Gas/Kerosine | 3 | 4 | 0.894(0.183–4.364) | 0.886(0.150–5.240) |
| Others (*d*) | 3 | 11 | 0.325(0.082–1.291) | 0.289(0.061–1.371) |
| **Habit of open defecation** | | | | |
| Yes | 27 | 85 | 0.523(0.327–0.836) | **0.508(0.301–0.857)\*** |
| No | 189 | 311 | Ref. | Ref. |
| **Habit of visiting traditional healers** | | | | |
| Yes | 71 | 99 | 1.469(1.021–2.113) | 1.362(0.904–2.052) |

*(Continued)*

**Table 7.** (Continued)

| Variables | Prevention Practice towards CL | | COR (95% CI) | AOR (95% CI) |
|---|---|---|---|---|
| | Good | Poor | | |
| No | 145 | 297 | Ref. | Ref. |
| **Knowing someone with CL** | | | | |
| Yes | 185 | 304 | Ref. | Ref. |
| No | 31 | 92 | 0.554(0.354–0.865) | **0.580(0.355–0.948)\*** |
| **Knowledge About CL** | | | | |
| Poor | 102 | 219 | 0.723(0.519–1.008) | 0.783(0.513–1.194) |
| Good | 114 | 177 | Ref. | Ref. |

Ref. = Reference \* = Significant association at P-value < 0.05, Hosmer and Lemeshow goodness of fit test (P = 0.248)

**AOR** = Adjusted Odds Ratio, **COR** = Crude Odds Ratio

**Others** a = Carpenter [4], Daily laborer [4], Security guard [2], Driver [1], Hairdresser [1] and Mason [1]

b = Block [2] and Stone [1]

c = Stone [2]

d = Dung [6], Manure [5] and Muck [3]

**Table 8. Characteristics of an in-depth interview participants, Kutaber district, Northeast Ethiopia, July 1 to August 15, 2022.**

| Participants | Gender | Age | Marital Status | Occupation | Place of Residence | Lesion/Scar Location | Involved As: |
|---|---|---|---|---|---|---|---|
| 1 | Female | 31 | Married | Housewife | Urban | Face | **Participants who have experienced the disease CL** |
| 2 | Female | 30 | Married | Merchant | Urban | Face | |
| 3 | Female | 19 | Single | Student | Rural | Face | |
| 4 | Male | 27 | Single | Unemployed | Rural | Face/Nose | |
| 5 | Male | 18 | Single | Student | Urban | Face/Chin | |
| 6 | Female | 18 | Single | Student | Urban | Face | |
| 7 | Male | 31 | Married | Private factory employee | Rural | Arm | |
| 8 | Female | 27 | Married | Teacher | Urban | Face | |
| 9 | Male | 34 | Married | Merchant | Rural | Face/Nose | |
| 10 | Male | 25 | Single | Religious Teacher | Urban | Face | |
| 11 | Male | 30 | Married | Civil servant | Rural | Arm | |
| 12 | Male | 56 | Married | District NTD Officer | Urban | - | **Key informants** |
| 13 | Male | 38 | Married | Dermatologist (at Hospital) | Urban | - | |
| 14 | Male | 30 | Single | Head of Health Center | Rural | - | |
| 15 | Male | 35 | Married | NTD focal person of HC | Urban | - | |
| 16 | Female | 24 | Single | Head of Health Center | Rural | - | |
| 17 | Female | 35 | Married | HEW | Rural | - | |
| 18 | Male | 52 | Married | Farmer (Community Elder) | Rural | - | |
| 19 | Male | 30 | Married | Head of Health Center | Rural | - | |
| 20 | Male | 35 | Single | Zone NTD Officer | Urban | - | |

HEW = Health Extension Worker, NTD = Neglected Tropical Disease

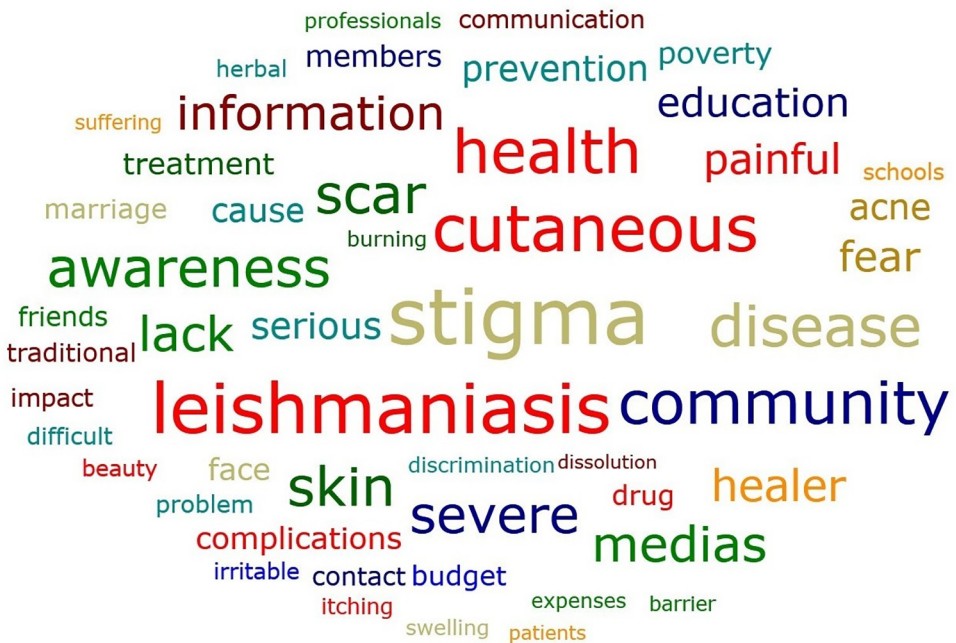

**Fig 2. Word cloud: 50 most occurring words showing "stigma" as the most frequent one.**

possible to move freely. There was a time that I isolated myself for not to have contact with the community.' A 52-year—male rural community elder stated that 'the lesion didn't smell bad and has no pus but they got ashamed of it and isolate themselves from the community.'

A 38-year dermatologist stated self-isolation experienced among the victims of CL in the community as: '. . .There are two patients that I know still. . . . Especially the male can't go to wedding house. Even if he goes, it is not to upset or to annoy his family and he returns without eating food. The female one is a grade 12 student. Her nose has been pierced and we have constructed it artificially and she has been healed from the disease. She entirely covered her face right now. She said to us, "Even though she is Muslim, she was not covered her face before and she fully covered her face as a result of the disease now.

**Dissolution of marriage.**   According to some participants, CL causes divorce and dissolution of marriage as a result of some cultural thinking within the community. A 19-year rural student stated, 'Cutaneous leishmaniasis causes divorce by itself. There is a saying that it needs protection (sexual avoidance or abstention) until getting healed from the disease and this is a cause for the dissolution of marriage.'

**Fear of disease complication.**   Victims of the disease stated that they fear further complications and additional damage to their bodies. A 30-year female married merchant stated, '. . . My body gets swollen. My face was changed and it itched me. I was psychologically affected as I think it might seriously destroy my body.' A 19-year rural student stated that 'I feel as I will not heal from this disease. Because when I see others, it gets swollen and is easy to be peeled/removed; mine is dried there.' In addition to this, a 27-year single rural victim stated, 'I was thinking it is going to pierce and entirely damage my nose, and beyond that, it looks forward to losing my sight.'

**Fear of future re-infection.**   Some of the victims were in fear of future re-infection. '. . . I would not be very happy as I get cured from CL because I am in fear of whether it will infect me once more or not;' (A 27- year unemployed rural victim). According to the participants, the

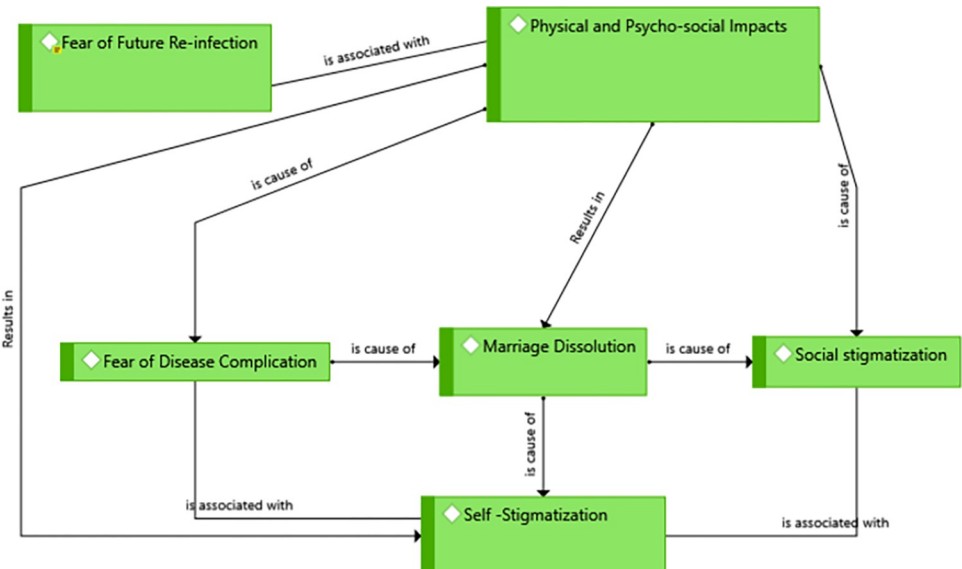

**Fig 3. A theme showing physical and psycho-social impacts of CL in Kutaber district, Northeast Ethiopia, 2022.**

major reason for this fear was that they didn't have awareness of the cause, prevention, and protective measures of the disease which in turn was supported by limited efforts of healthcare faculties and health professionals in creating awareness for the community as well as the inability of media to disseminate the information (**Fig 3**).

### Theme 2: treatment experiences of the community

**Types of treatments used.**   Different treatment methods were experienced in the community according to the informants. These treatment methods were traditional treatment, modern treatment through health care facilities, home treatment and religious treatment. The most widely practiced treatment type was traditional treatment.

**Treatment method preference.**   Participants of the study stated priority was given to traditional over modern treatment due to different reasons such as: thinking the disease was untreatable in modern treatment, thinking that CL heals with traditional treatment, lack of information about the presence of its modern treatment, thinking of faster healing time in traditional treatment, education, and awareness gap, minimum/low cost of treatment from the traditional treatment, the requirement of longtime stay at modern treatment facilities, fearing of pain as a result of the injection, inability to understand the harmful treatment outcomes following traditional treatment, the belief that CL has no modern treatment being in a remote area and long distance to the modern treatment center. A 27-year male rural resident stated, *'I don't believe there is anyone visiting Boru-Meda hospital for treatment without visiting traditional healer/s at first. Everyone visits the hospital after he/she tried and gets tired.'* Similarly, A 38-year married dermatologist stated, *'. . . may be patients from the town, closer to health extension workers and who are educated are the only ones coming without touching it with traditional and home-made treatment.*

Visiting one or more traditional healers was very commonly experienced in the community after having the disease according to the study participants. A 27- year- old male unemployed rural victim stated, *'I have visited many areas for traditional treatment. (. . .) After that, I went to another healer around the Boru-Selassie area having different medicine than the first one*

(. . .). *I visited another traditional healer for the third time and that was also not effective for me and it doesn't help me to heal from cutaneous leishmaniasis.'* A 25-year religious teacher stated, '*There was one traditional healer in the rural area of the district called Kundi. She has her herb. She gives that herb to the community and the community uses it.'* A 19-year urban student also said that '*. . . I found traditional healers and I tried with it'* (**Fig 4**).

### Theme 3: information dissemination and communication experiences

**Information dissemination.**  According to the participants, there was no formal information disseminated for the community regarding CL. Mostly, this was resulted from: limited efforts of healthcare facilities and schools, lack of medias disseminating the information, limited efforts of health professionals and low attention given from the government. A 27-year-old male victim stated, '*No medias are available for cutaneous leishmaniasis information dissemination.' . . . There is information about diseases like trachoma, hypertension, diabetes mellitus, TB and cancer delivered via radio, television and FM radios. There is a program called health at home for such diseases and I listened it. But there is nothing about cutaneous leishmaniasis.'*

**Community as an information hub.**  According to the participants, even though the correctness of the information is questionable, the community was a center of information about CL and communication through informal way was commonly experienced. Hearing about the disease is common in the community rather than the health facility.

In addition to media, there were another barrier of information dissemination and awareness creation about CL in the community according to the participants. The most frequently explored barriers were: limited involvement of partners, limited efforts of healthcare facilities and schools, limited efforts of health professionals (**Fig 5**).

### Theme 4: prevention experience of the community

The community has done nothing or very little to prevent the disease CL according to the study participants. A 30- year female merchant stated, '*I see the community nothing to do.'* A 19- year male student stated, '*. . . we used nothing for prevention.'* Similarly, an 18- year female student stated, . . . '*They have never used personal protective equipment (PPEs)'*

According to the study participants, the community had an experience of fetching water, visiting forests, and hunting rock hyraxes (the reservoir host of the agent) mainly without

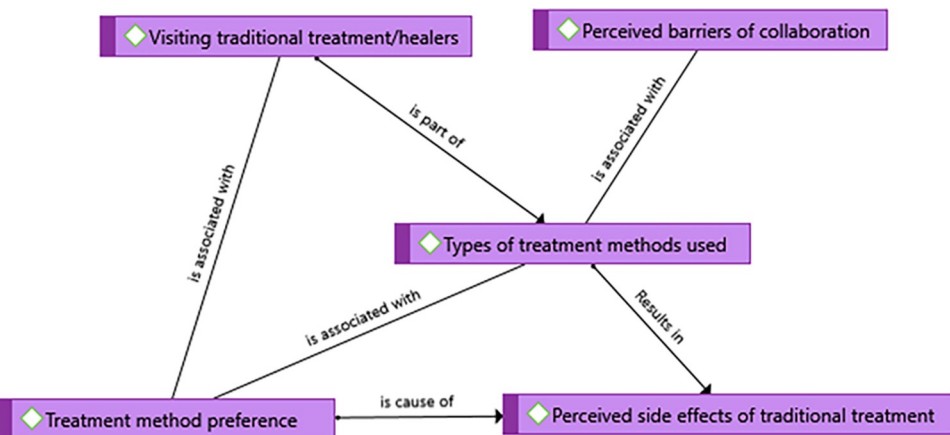

**Fig 4. A theme showing treatment experiences of the community in Kutaber district, Northeast Ethiopia, 2022.**

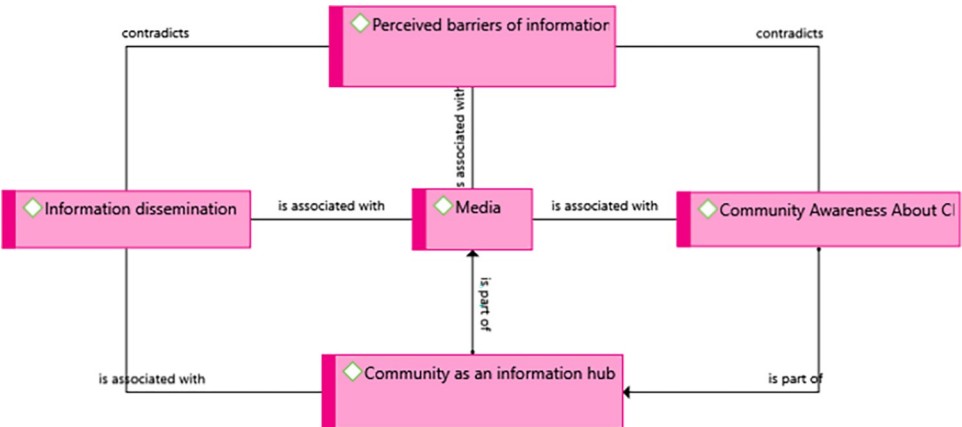

**Fig 5. A theme showing information dissemination and communication experiences of the community in Kutaber district, Northeast Ethiopia, 2022.**

using appropriate PPEs and at the time suitable to the sandfly bite. A 31-year male private factory employee stated, *'They go to cut bushes to fence their crops, to see their crops and grasses nearby the forest. . . . for hunting; sometimes they go with their dogs. Mainly they go to the forest in the early morning/dusk time. . . . People hunt hyraxes for thin persons as medicine. Most of the time, they are going to hunt them in the morning.'* A 19-year female rural student stated, *'Most of the time we fetched water in the morning and night'*

### Perceived barriers to implementing prevention against CL

There are barriers to implementing prevention against the disease that are frequently stated by the study participants. These are lack/gap of awareness about CL, the problem of information delivery, the inability of following up on the treatment, lack of prevention equipment, low level of attention given from the government, and limited efforts of health care facilities and professionals. A 56-year district NTD officer stated, *'It is due to the low level of attention given from the government like other diseases regarding prevention and surveillance. (. . .) limitation of creating awareness for the community from health care facilities and professionals such as health extension workers.'* A 27- year male rural victim stated, *'The obstacle to prevent the disease in the community is. . . The other problem is the awareness gap in the community about the transmission, prevention, and cause of cutaneous leishmaniasis'* (**Fig 6**).

### Theme 5: recommendations of study participants

According to the study participants, health professionals, medias, schools and healthcare facilities were recommended to do their best for the community against this disease (**Fig 7**).

### Discussion

This study found that the proportion of good knowledge about CL was 47.5% (95% CI: 43.5–51.6%) and this indicates the residents' knowledge about the disease was poor. This is supported by the qualitative part of this study; in which a 27- year- old male rural participant stated *'I don't know the cause, prevention and protective measures of cutaneous leishmaniasis.'* A 52-year-old rural community elder also stated, *'It is token that its cause was bats but we don't know the cause of the disease clearly, whether it is from tree, air or heat.'* This finding is higher than the finding of a

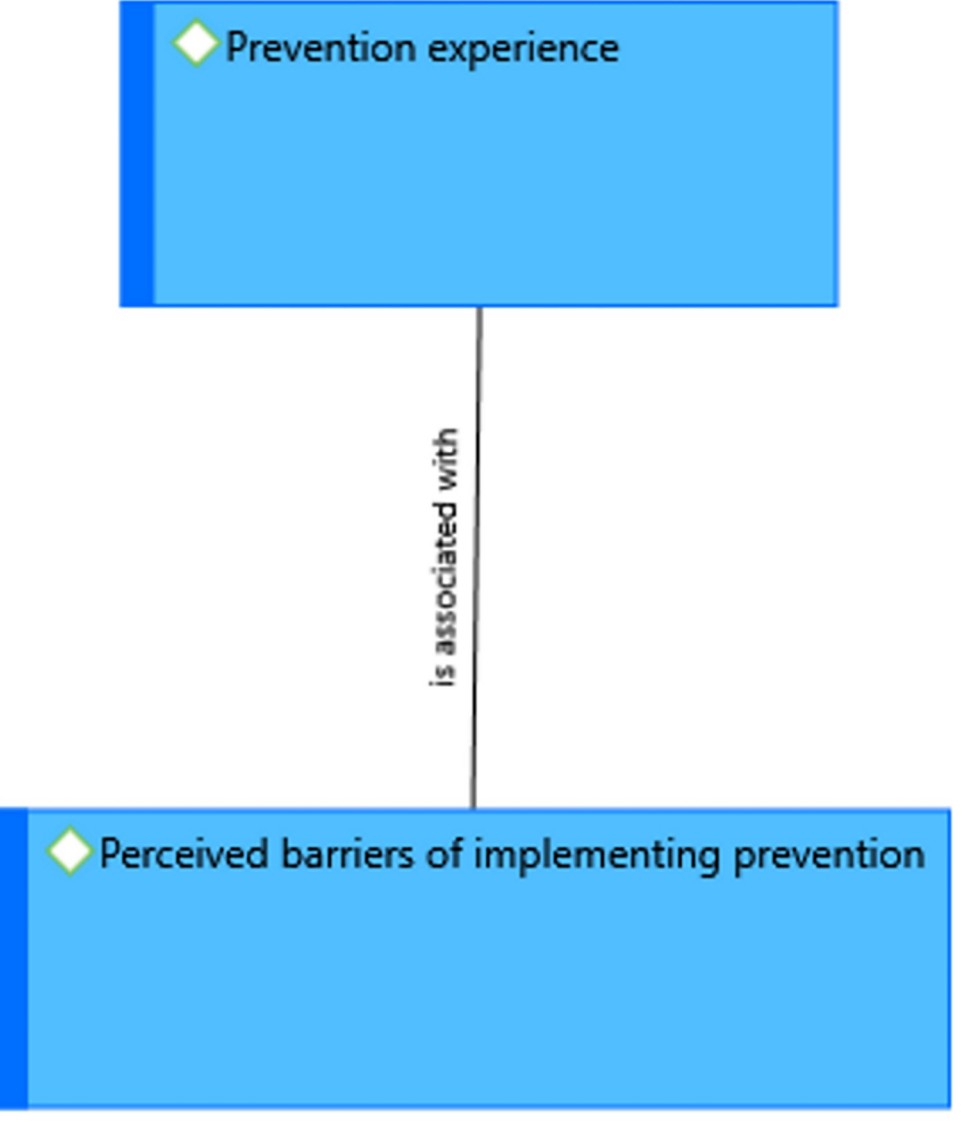

**Fig 6. A theme showing prevention experiences of the community in Kutaber district, Northeast Ethiopia, 2022.**

study conducted in Southwestern Yemen, which was 22.3% [25]. The possible reason might be due to the major collapse of the healthcare infrastructure following the ongoing war, which resulted in the interruption of health education and health information dissemination and the study time difference. This finding is lower than the findings of the studies conducted in Kenya (89.4%) [26], Sodo district, Southern Ethiopia (61.9%) [21], Ochello, Gamo-Gofa Zone, Southern Ethiopia (67.6%) [27], and Ganta-afeshum district, Tigray (84%) [28]. The discrepancies might be due to differences in the sample size of the studies, study period, the endemicity level of the disease in the areas, and levels of information communication.

This study revealed that, residents who were unable to read and write and able to read and write only had lower odds of having good knowledge about CL as compared to those residents who have completed secondary education and above. This finding is inconsistent with a study conducted in the Sodo district, South Ethiopia in which residents who were unable to read and write and those who were able to read and write only had no significant difference to have

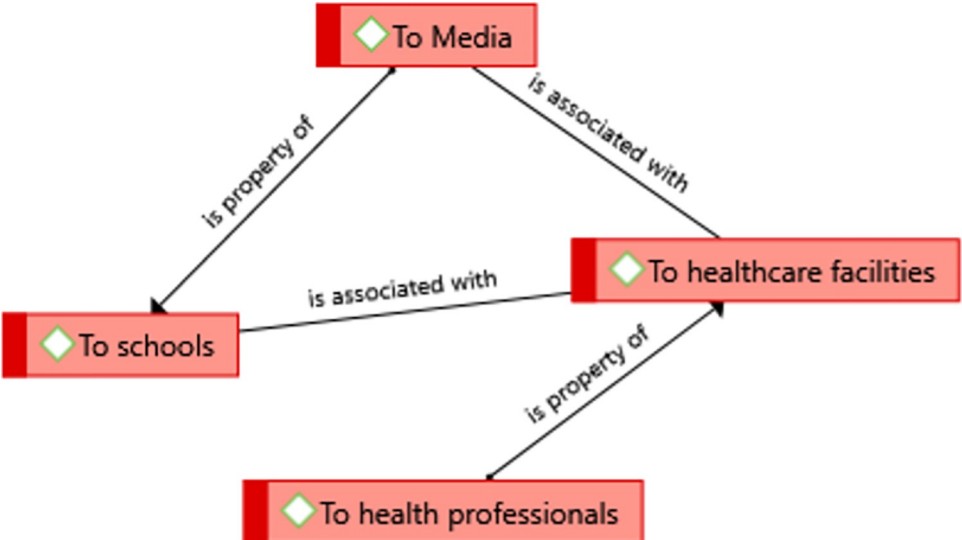

**Fig 7. : A theme showing recommendations of the study participants in Kutaber district, Northeast Ethiopia, 2022.**

good knowledge about CL as compared to those residents who were in secondary education and above [21]. This might be due to the differences in the level of health education provision, information communication, and study time.

Residents who were in the poorest household wealth had lower odds of having good knowledge about CL as compared to those residents who were in the richest household wealth. This is supported by a study conducted in Saudi Arabia which demonstrated a significantly higher knowledge score correlated with higher family income [29]. The possible reason for this might be: residents in the poorest household wealth have limited access to media and schools which resulted in the missing of disseminated health information and provision of education to residents in the richest household wealth.

The finding of this study revealed that the proportion of a positive attitude towards CL was 54.1% (95% CI: 50.0–58.1%). This is in line with the findings of the studies conducted in Kenya with 52.4% [26] and Sodo district in Southern Ethiopia with 53.4% [21]. This consistency might be due to the similarity of the study design. But the finding obtained from this study is higher than a study conducted in Southwestern Yemen, which was 24.4 [25]. This finding is lower than the finding of a study conducted in Iran, which was 71.6% [24]. This inconsistency might be due to differences in socio-cultural settings, study time, sample size, and level of information dissemination and health education provision.

Residents aged >54.5 years had lower odds to have a positive attitude towards CL as compared to those residents aged between 18–24.5 years. This is not agreed with a study conducted in the Sodo district of southern Ethiopia; where there was no significant difference among residents aged > 54.5 years and that aged between 18–24.5 years in having a positive attitude towards CL [21]. This disagreement might be due to the difference in the level of information dissemination and health education provision in the study areas.

The finding of this study revealed that the proportion of a good prevention practice towards CL was found to be 35.3% (95% CI: 31.5–39.2%). This indicated that the prevention practice of residents' is poor. This is similarly indicated by the qualitative part of this study in which; A 30- year female merchant stated, '*I see the community nothing to do for prevention.*' A 19- year male student also stated, '*. . . we used nothing for prevention.*' Similarly, an 18- year female

student stated, . . . *'they have never used PPEs'* This finding is consistent with the finding of the study conducted in Ochello, Gamo-Gofa Zone of South Ethiopia, which was 37.5% [27]. The consistency might be due to the similarity of the study design. The finding is higher than the finding of a study conducted in Ganta-afeshum district, Tigray in which 18% of the respondents had a good attitude [28] and lower than the findings of the studies conducted in Mount Elgon focus of Kenya, which was 57.9% [26] and Sodo district, Southern Ethiopia, which was 50.4% [21]. This discrepancy might be due to the differences in study time, sample size, level of health education provision, and knowledge level of the community about CL. Males had higher odds to have a good prevention practice as compared to females.

Residents aged between 24.5–34.5 years and 34.5–44.5 years had lower odds of having a good prevention practice towards CL as compared to those who were aged between 18–24.5 years. This might be due to the major tasks performed by these age groups and removing of some preventive materials following inconveniences in performing their tasks (for example, worn-out of long sleeve clothes). Residents who were in the poorest, poorer, and middle household wealth had lower odds to have a good prevention practice towards CL as compared to those residents who were in the richest household wealth. This is agreed with a study conducted in Colombia in which households with higher economic status had access and practiced to more costly/ effective preventive measures [30].

Participants' fear was related to social stigmatization, the associated pain, failure in healing, disease complication, disability, permanent scar, and future re-infection. Women were more affected by the aesthetic and disfigurement aspects of the scars. This was also clearly observed in the finding of a study conducted in Tunisia [31]. Even though CL is a very old disease in the Kutaber district, the lack of knowledge of CL victims of their disease was a big issue, creating additional stress for the victims regarding disease complications and future re-infections. Social stigmatization was the most frequent experience that the victims had faced including divorce and dissolution of marriage following the disease. This finding is consistent with studies conducted in Morocco [32] and Yemen [33].

This study explored, social stigmatization was frequently linked to CL for different reasons such as the community's misconceptions of the disease such as transmission from person-to-person contact, through direct observation, via food, through houseflies, and bats urine. This misconception was also shown by other studies in different areas [16,25,34,35]. This study also explored the treatment experiences of victims. The experienced traditional treatment side effects like disease complication, delays of healing, and scar formation were very striking. This is consistent with a study conducted in Morocco [32]. This study found that the community had no adequate awareness of CL regarding its cause, transmission, and prevention methods. This is in line with a qualitative study conducted in Iran where participants described how they were unaware of the community and saw how this was one of the important factors increasing their suffering [4].

## Conclusion

Even though a positive attitude was registered among residents of Kutaber district, the level of overall knowledge and prevention practice towards CL was quite poor. Educational status of residents, years of residence, household wealth, use of media, and knowing someone with CL were factors significantly associated with knowledge about CL. Age of residents, years of residence, and habit of visiting traditional healers were factors significantly associated with attitude towards CL. Gender of residents, age of residents, household wealth, condition of the wall surface of the house, location of the house from creeks/waterways, the habit of open defecation, and knowing someone with CL were factors significantly associated with prevention practice towards CL.

CL is a true problem in the district and is very much intertwined with physical and psychosocial effects such as pain, scar formation, and social and self-stigmatization. Regarding treatment methods, due to limited efforts of healthcare professionals, limited roles played by the media, budget problems, low attention given from the government, lack of information dissemination, and lack of awareness, patients of the disease preferred to visit traditional treatment methods over the modern treatment and experienced its side effects following the treatment. The finding of this study implied that there is a need for intervention in terms of rigorous provision of health education to the residents of the district about the cause, transmission mechanisms, the vector and its breeding site, the intermediate host, and treatment mechanisms of the disease with a strong integrated effort from healthcare providers, government officials, and non-governmental organizations (NGOs). Moreover, working on the prevention strategies such as avoid working outdoors between dusk and dawn, wearing protective clothing and insect repellant use should be rigorously implemented.

## Limitations of the study

The study was done with some limitations. Due to the interviewer's administration of the questions, some variables may be subject to social desirability bias. Recall bias may be an issue because of the examination of self-reported behavior patterns was retroactive. Moreover, as participants from six different Kebeles completed the surveys, the comparison between them was not made to understand if CL perceptions were conserved across the households, or whether there were certain foci that held strongly to different knowledge, attitudes, prevention practices. In addition to these, in dichotomizing the outcome variables (i.e., knowledge, attitude and prevention practice), the difference between those near the mean score might be very narrow and its accuracy might be compromised as a result.

## Supporting information

**S1 Questionnaire. English version of the questionnaire.**
(DOCX)

**S1 Datasets. Datasets for KAP towards CL among Kutaber district residents.**
(SAV)

## Acknowledgments

We deeply express our appreciation to Amhara regional health bureau, Kutaber district health office, Boru-Meda Hospital, Kebele leaders, professionals involved in an expert judgment, study participants, data collectors, and other individuals or organizations that have participated in the study directly or indirectly.

## Author Contributions

**Conceptualization:** Abebe Kassa Geto, Asmamaw Malede.

**Data curation:** Abebe Kassa Geto, Mistir Lingerew, Leykun Berhanu, Metadel Adane.

**Formal analysis:** Abebe Kassa Geto, Asmamaw Malede, Mistir Lingerew, Ayechew Ademas, Leykun Berhanu, Genanew Mulugeta Kassaw, Belachew Tekleyohannes Wogayehu, Metadel Adane.

**Funding acquisition:** Abebe Kassa Geto.

**Investigation:** Abebe Kassa Geto, Ayechew Ademas, Leykun Berhanu, Genanew Mulugeta Kassaw, Metadel Adane.

**Methodology:** Abebe Kassa Geto, Mistir Lingerew, Gete Berihun, Ayechew Ademas, Leykun Berhanu, Genanew Mulugeta Kassaw, Metadel Adane.

**Project administration:** Abebe Kassa Geto, Gete Berihun, Genanew Mulugeta Kassaw, Metadel Adane.

**Resources:** Abebe Kassa Geto.

**Software:** Abebe Kassa Geto, Asmamaw Malede, Gete Berihun, Genanew Mulugeta Kassaw, Belachew Tekleyohannes Wogayehu, Metadel Adane.

**Supervision:** Abebe Kassa Geto, Asmamaw Malede, Alebachew Bitew Abie, Ayechew Ademas, Genanew Mulugeta Kassaw, Belachew Tekleyohannes Wogayehu, Metadel Adane.

**Validation:** Abebe Kassa Geto, Alebachew Bitew Abie, Ayechew Ademas, Genanew Mulugeta Kassaw, Belachew Tekleyohannes Wogayehu, Metadel Adane.

**Visualization:** Abebe Kassa Geto, Alebachew Bitew Abie, Genanew Mulugeta Kassaw, Belachew Tekleyohannes Wogayehu, Metadel Adane.

**Writing – original draft:** Abebe Kassa Geto, Mistir Lingerew.

**Writing – review & editing:** Abebe Kassa Geto, Asmamaw Malede, Mistir Lingerew, Alebachew Bitew Abie, Gete Berihun, Ayechew Ademas, Leykun Berhanu, Genanew Mulugeta Kassaw, Belachew Tekleyohannes Wogayehu, Metadel Adane.

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
