## [Decision Letter · Decision Letter 0]

21 Jan 2024

Dear Mr. Geto,

Thank you very much for submitting your manuscript "Date: 18/06/2023

 Knowledge, attitude, prevention practice and lived experience towards cutaneous leishmaniasis and associated factors among residents of Kutaber district, Northeast Ethiopia, 2022: a mixed method study" for consideration at PLOS Neglected Tropical Diseases. As with all papers reviewed by the journal, your manuscript was reviewed by members of the editorial board and by several independent reviewers. In light of the reviews (below this email), we would like to invite the resubmission of a significantly-revised version that takes into account the reviewers' comments. 

We cannot make any decision about publication until we have seen the revised manuscript and your response to the reviewers' comments. Your revised manuscript is also likely to be sent to reviewers for further evaluation.

Sincerely,

Álvaro Acosta-Serrano

Section Editor

Reviewer's Responses to Questions

**Key Review Criteria Required for Acceptance?**

**Methods**

-Are the objectives of the study clearly articulated with a clear testable hypothesis stated?

-Is the study design appropriate to address the stated objectives?

-Is the population clearly described and appropriate for the hypothesis being tested?

-Is the sample size sufficient to ensure adequate power to address the hypothesis being tested?

-Were correct statistical analysis used to support conclusions?

-Are there concerns about ethical or regulatory requirements being met?

Reviewer #1: Methods are well designed to answer the questions raised by the study.

My concern is about the implementation of the questionaires. It is not clear if the questions were adapted to the cultural and social aspects of the communities involved. In my opinion this is of upmost importance to get relliable information. The authors should explain if it was done and how. In case this adaptation was not done, authors should discuss this important limitation.

Reviewer #2: This manuscript addresses an interesting and valuable multifaceted question: how do people in Northeast Ethiopia understand, cope with, treat and prevent cutaneous leishmaniasis in 2022? The authors have compiled a vast amount of data, both quantitative and qualitative, and from this, have drawn conclusions according to the responses of 612 survey respondents and from 20 in depth interviews with patients and key informants. Although the manuscript contains a wealth of information, the section organisation, data presentation and validation, and deeper data interpretation is lacking. Consequently, to publish this work with PLoS NTD will require a major rewrite. Below are some of the areas that need additional revision and I have introduced several comments that, if addressed, will make it easier for future readers to understand the significance of this work. There are no regulatory or ethical concerns.

**Results**

-Does the analysis presented match the analysis plan?

-Are the results clearly and completely presented?

-Are the figures (Tables, Images) of sufficient quality for clarity?

Reviewer #1: The analysis and results seems clear presented.

Reviewer #2: The analysis of the survey results are highly confusing and where survey questions were repeated, it is not clear why different responses were recorded. The tables, as they are presented are fractionated and more applicable for a thesis style of writing. It would be far more informative to have all the responses to each question in a single table, with colour coded sections to represent each section. Suggest also revisiting all the data as the text does not match what is in the tables, and some of the tabulated data do not agree when it should.

ie. 

Comments on Figures

Fig 1. 

a) Label the different maps in the order you want the reader to look at them. By default, an English reader will read from left to right so will naturally look at the blue map first and then the pink one, which is incorrect if the point is to move from larger to smaller geographical scale.

b) For the first blue panel, please add in the many countries bordering Ethiopia so it is easier to locate the study region and understand its position on the continent. Include a marker for Addis as you mention it in your M&M.

c) Add in map scale to the other 3 panels

d) If you highlight the Kataber District on the orange Amhara Region panel, the pink map panel can be removed.

e) It is more informative to add in some geographical features to the Kutaber Woreda map such as rivers, lakes, major cities (Kutaber, Dessie etc), deserts, mountains, and primary roadways

Fig 3. 

On my printout, this figure is impossible to read.

a) Please ensure everything has been standardised to the same quality, size and font. 

b) change black lettering on dark green or purple bottom squares to a white font

c) need to incorporate directionality to the image = what do you want the reader to read first, second etc...

d) the font size on the branches / lines must be increased or if not enough room, label each line with a number and describe it in the figure legend

**Conclusions**

-Are the conclusions supported by the data presented?

-Are the limitations of analysis clearly described?

-Do the authors discuss how these data can be helpful to advance our understanding of the topic under study?

-Is public health relevance addressed?

Reviewer #1: The conclusions may be biased by a possible lack of cultural and social integration of the tools used to evaluate some factors that were investigated. Authors should discuss this limitation and possible interference on the interpretation of the data collected.

Reviewer #2: All of these conclusions need to be revisited by the authors. The data requires correction, the study details need to be expanded (ie. what were the exclusion / inclusion criteria for the participants?), what knowledge gap is this data filing and how will it be used in the future, how will this study improve the leishmaniasis knowledge, treatment, prevention of the residents of Kutaber district?

**Editorial and Data Presentation Modifications?**

Reviewer #1: Minor revision according to the comments already done.

Reviewer #2: Below are comments / suggestions / questions on specific sections of the manuscript that, if addressed, will further enhance value, accuracy and clarity of this manuscript.

Abstract:

Rephrase the Results section to be more compelling with better clarity (not necessary to report CI's in abstract. ie.

"The survey respondents in Kutaber district showed good knowledge of (47.5%), a positive attitude towards (54.1%) and good prevention practices (35.3%) regarding cutaneous leishmaniasis"

It would be more informative if definitions for what good and positive mean. If 50% was a pass, then 47.5% and 35.3% would not be classified as "good". What is a positive attitude - is it positive because the value exceeded 50%?

Introduction:

Should logically flow from a global scene to something specific to this research. line 53 starts to discuss Ethiopia and then by line 57 jumps back to the global situation of CL.

L53: "CL has been known since 1913" - too vague. was known by who? local populations? was it introduced as a disease in 1913 or has been endemic for much longer?

L68: explain why only the highland areas are endemic - is there something special about the highland regions (climate/geography/flora/fauna/how people live) or is it simply there is no data from other regions to compare to? Should discuss the cycle of leish in this region, especially as your survey questions mention rock hyrax and bats.

L77:what were the gaps in prevention methods reported? what prevention methods were used?

L78: why is CL a growing concern in the Amhara region?

L81, 82, 83 and elsewhere - be aware of the overuse of the word "conducted"

L87: why is it important to capture "lived experience" and what do you mean by this?

L88: what will all this data be used for / contribute to in the future? is there evidence from other CL areas or other diseases in Ethiopia that such knowledge is important?

M and M:

L91: explain in detail why the Kutaber district was selected as the study site for this work

L105: those "who have experienced CL" means of family or personally? in what time frame? 

L106: what are the qualifications of a key informant?

L107: what were the inclusion and exclusion criteria used to define the study populations for both the quantitative and qualitative studies

L124: considerable CL cases within what time frame? how were the cases documented - via the community, healers or hospital records?

L157 - 1179: explain why the scoring was based on the mean scores. was this overall or per the sampling sites Kutaber, Alansha, Doshign, Beshilo, Haroye and Kundi?

L173: it is not enough to simply know bednet utiilization - what type of bednets did people have access to? many bednets are for malaria prevention and consequently the weave is not small enough to prevent sand flies from biting. the lack of using a bednet, regardless of the presence of holes, for CL could indicate the person knows it cannot protect against biting sand flies! 

L174: what is house spray?

L174: what CL control and treatment options were available in this district?

L190: further describe the parameters used to assess household wealth and explain why some of those parameters were not used in Kutaber

L198: what language was used for the questionnaire? were translators used?

L219: where is the data stored (including signed informed consent sheets) and how long will data be stored before it is destroyed?

Results:

L264: explain what different types of media you assessed the usage of for scoring

L266: "knew someone with CL" - at the time of the survey this was someone with an active case or rather at some point in the interviewee's life, they had encountered someone with CL?

L275: explain what prompted the questionnaire question of CL transmission through bat urine? is it because of caves in the area? or behaviours? or a legend?

Table 2: Question: Is CL a communicable disease? Communicable is considered a health jargon word. How did you confirm that persons answering this questions understood what it meant in the way you wanted them to?

L280 onwards: It is troubling to see that the numbers described in "attitude towards CL" section do not match what is presented in the tables.

ie. "About 378 of the study particpants agreed the CL is a health problem in their area, but about 75 of the participants disagreed with this idea". According to the referenced Table 3, only 278 patients agreed or strongly agreed with this statement and 230 disagreed with this idea. 

This mismatch between table and text is throughout this section and other places in the results.

L301: participants prefeed modern medicine over what?

Some sections of Table 5 and 6 are not in logical agreement. Considering the same participants are completing the survey at a single timepoint, it is not explained why answers differ between the two tables when certain stats are recorded like years of residence. Are the tables showing that how long a person lives in this district affects their knowledge of CL (Table 5) or their attitude towards CL (Table 6) in opposite ways (especially if participants have only been there for 1 -2 years? This is interesting as it shows that knowledge of CL is poor in this group yet their attitude towards CL can remain positive? This and other such observations need to be further expanded in the discussion.

L365 (section start): Expand on what metrics you used to assess physical, social, psychological impacts and whether body language / speech / emotional inputs were also assessed during interviews.

Note: it would be helpful for the reader to understand what some of the quotations are referring to ie. what does a "cut nose" mean? Is this a treatment?

Such powerful quotes you have captured!

L441: priority was given to traditional healers. How did this compare to the data collected from the questionnaire?

Discussion: Consider tabulating the prioritized data you have mentioned so the discussion is not re-iterating data presented in the results and rather explain why these trends are appearing and what this means for future disease management.

L588: What are the traditional treatment side effects?

Why do more people continue to seek out a healer when they see that friends/family are not being healed, especially if there are side effects to this treatment? Is it simply a fear of injections that drives this behaviour or something else?

**Summary and General Comments**

Reviewer #1: No further comments.

Reviewer #2: As outlined in the previous sections, I have suggested areas that need further clarification, information or correction. With a serious rewrite, this manuscript will provide valuable information to decision makers in public health, as to where resources for CL should be prioritised - ie. prevention, education or treatment. The quotes captured during the in depth interviews is insightful and clearly shows how keen those living in this district are to better understand and prevent CL. The recommendation of major revision does not require new experiments, but rather a better analysis of the data. As participants from six different Kebeles completed the surveys, there should be a comparison between them (or a strong explanation why not), to understand if CL perceptions were conserved across the households, or whether there were certain foci that held strongly to different knowledge, attitudes, practices.

PLOS authors have the option to publish the peer review history of their article (what does this mean?). If published, this will include your full peer review and any attached files.

Reviewer #1: Yes: Paulo R. L. Machado

Reviewer #2: No
---

## [Decision Letter · Decision Letter 1]

18 Jul 2024

Dear Mr. Geto,

Thank you very much for submitting your manuscript "Knowledge, attitude, prevention practice and lived experience towards cutaneous leishmaniasis and associated factors among residents of Kutaber district, Northeast Ethiopia, 2022: a mixed method study" for consideration at PLOS Neglected Tropical Diseases. As with all papers reviewed by the journal, your manuscript was reviewed by members of the editorial board and by several independent reviewers. The reviewers appreciated the attention to an important topic. Based on the reviews, we are likely to accept this manuscript for publication, providing that you modify the manuscript according to the review recommendations. 

Note from editor: the manuscript has significantly improved, but please address all issues clearly described by both reviewers. Some of these were already pointed out by the reviewers and not properly addressed in the revised version.

Sincerely,

Álvaro Acosta-Serrano

Section Editor

Reviewer's Responses to Questions

**Key Review Criteria Required for Acceptance?**

**Methods**

-Are the objectives of the study clearly articulated with a clear testable hypothesis stated?

-Is the study design appropriate to address the stated objectives?

-Is the population clearly described and appropriate for the hypothesis being tested?

-Is the sample size sufficient to ensure adequate power to address the hypothesis being tested?

-Were correct statistical analysis used to support conclusions?

-Are there concerns about ethical or regulatory requirements being met?

Reviewer #1: Very well done, methodology seems adequate and well designed to obtain fine data and interpretation.

However, the outputs "Good Knowledge versus Poor Knowledge", "Attitude" and "Prevention Practice" should not be dihcotomic. The difference between those near the mean score is very narrow, its accuracy may be lower than desirable in a so complex reality. I think that it is important to modify it, and establish a third output, for example: "Regular Knowledge" (also for the others). Another option would be exclude from the output those who are near the mean score (for example: 5% above or low?).

Reviewer #2: Yes

**Results**

-Does the analysis presented match the analysis plan?

-Are the results clearly and completely presented?

-Are the figures (Tables, Images) of sufficient quality for clarity?

Reviewer #1: Results are well described and analyzed. Please see observations above.

Reviewer #2: Yes although the results become extremely monotonous and will be difficult for a reader to stay engaged. 

I would like to suggest that three of the rebuttal comments are embedded into the manuscript as they give valuable added insight to the reader.

 Q. Explain why only the highland areas are endemic - is there something special about the highland

regions (climate/geography/flora/fauna/how people live) or is it simply there is no data from other regions to compare to? Should discuss the cycle of leish in this region, especially as your survey questions mention rock hyrax and bats.

A. Thank you. This is because the intermediate hosts (rock hyraxes), the species of the disease vector

(sandfly species) are commonly found and intertwined in the dense-forested highland areas of the regions

Q. It is not enough to simply know bed net utilization - what type of bed nets did people have access to? many bed nets are for malaria prevention and consequently the weave is not small enough to prevent sand flies from biting. the lack of using a bed net, regardless of the presence of holes, for CL could indicate the person knows it cannot protect

against biting sand flies!

A. Regarding bed nets, there are evidences that stated the distribution of insecticide-treated net and house spraying to prevent malaria have a positive impact on the control of sand flies. WHAT ARE THESE EVIDENCES? REFERENCES?

Q. Explain what different types of media you assessed the usage of for scoring.

A. The types of medias we assessed for the usage were radio, television, microphone and social medias (Facebook, telegram, and YouTube).

**Conclusions**

-Are the conclusions supported by the data presented?

-Are the limitations of analysis clearly described?

-Do the authors discuss how these data can be helpful to advance our understanding of the topic under study?

-Is public health relevance addressed?

Reviewer #1: I agree with the conclusions and limitations presented. However a better analysis of the outputs "Good Knowledge versus Poor Knowledge", "Attitude" and "Prevention Practice" should be provided

Reviewer #2: In your conclusions, now that you have assessed the data, it would be highly valuable to add your own data-driven perspective on what would be the best specific practice moving forward as a follow on from your last sentence: "The finding of this study implied that there is a need for intervention in terms of rigorous provision of health education and implementation of prevention strategies". 

Expand on your ending sentence above: "improved health education" TO WHOM (healthcare workers, healers, communities, patients) and elaborate on what prevention STRATEGY WOULD BE THE MOST EFFECTIVE to prevent CL transmission in this region? If you want this data be influential, create signposts for those seeking impactful advice on CL management, whether it be community members, doctors, politicians, health sector.

**Editorial and Data Presentation Modifications?**

Reviewer #1: No further comments.

Reviewer #2: Changes to the manuscript are well received however there are a few items that need further attention as described.

**Summary and General Comments**

Reviewer #1: No further comments.

Reviewer #2: The authors have invested a great amount of time in responding to my suggestions and revising the manuscript. It has improved the context of the work and the quality of the assessment, and the background of CL in Ethiopia is better described.

PLOS authors have the option to publish the peer review history of their article (what does this mean?). If published, this will include your full peer review and any attached files.

Reviewer #1: No

Reviewer #2: No

Figure Files:

Data Requirements:

Reproducibility:

References

---

## [Editor Report · Decision Letter 2]

2 Aug 2024

Dear Mr. Geto,

We are pleased to inform you that your manuscript 'Knowledge, attitude, prevention practice and lived experience towards cutaneous leishmaniasis and associated factors among residents of Kutaber district, Northeast Ethiopia, 2022: a mixed method study' has been provisionally accepted for publication in PLOS Neglected Tropical Diseases.

Best regards,

Álvaro Acosta-Serrano

Section Editor

---

## [Editor Report · Acceptance letter]

16 Aug 2024

Dear Mr. Geto,

We are delighted to inform you that your manuscript, " Knowledge, attitude, prevention practice and lived experience towards cutaneous leishmaniasis and associated factors among residents of Kutaber district, Northeast Ethiopia, 2022: a mixed method study ," has been formally accepted for publication in PLOS Neglected Tropical Diseases.

Best regards,

Shaden Kamhawi

co-Editor-in-Chief

Paul Brindley

co-Editor-in-Chief
